# Attitude Determination for Unmanned Cooperative Navigation Swarm Based on Multivectors in Covisibility Graph

**Yilin Liu, Ruochen Liu, Ruihang Yu \*, Zhiming Xiong, Yan Guo, Shaokun Cai and Pengfei Jiang**

College of Mechatronics and Automation, National University of Defense Technology, Changsha 410073, China
\* Correspondence: yuruihang@nudt.edu.cn

**Abstract:** To reduce costs, an unmanned swarm usually consists of nodes with high-accuracy navigation sensors (HAN) and nodes with low-accuracy navigation sensors (LAN). Transmitting and fusing the navigation information obtained by HANs enables LANs to improve their positioning accuracy, which in general is called cooperative navigation (CN). In this method, the accuracy of relative observation between platforms in the swarm have dramatic effects on the positioning results. In the popular research, constructing constraints in three-dimensional (3D) frame could only optimize the position and velocity of LANs but neglected the attitude estimation so LANs cannot maintain a high attitude accuracy when utilizing navigation information obtained by sensors installed during maneuvers over long periods. Considering the performance of the inertial measurement unit (IMU) and other common sensors, this paper advances a new method to estimate the attitude of LANs in a swarm. Because the small unmanned nodes are strictly limited by relevant practical engineering problems such as size, weight and power, the method proposed could compensate for the attitude error caused by strapdown gyroscopic drift, which only use visual vectors built by the targets detected by cameras with the function of range finding. In our method, the coordinates of targets are mainly given by the You Only Look Once (YOLO) algorithm, then the visual vectors are built by connecting the targets in the covisibility graph of the nodes in the swarm. The attitude transformation matrices between each camera frame are calculated using the multivector attitude determination algorithm. Finally, we design an information filter (IF) to determine the attitude of LANs based on the observation of HANs. Considering the problem of positioning reference, the field test was conducted in the open air and we chose to use two-wheeled robots and one UAV to carry out the experiment. The results show that the relative attitude error between nodes is less than 4 degrees using the visual vector. After filtering, the attitude divergence of LANs' installed low precision IMU can be effectively constrained, and the high-precision attitude estimation in an unmanned CN swarm can be realized.

**Keywords:** cooperative navigation; Wahba's problem; attitude determination; covisibility graph



## 1. Introduction

With the rapid development of the robot industry, accurate and continuous positioning for robots has become particularly important. The error of low-cost inertial navigation systems (INS) accumulates quickly without global navigation satellite system (GNSS) assistance when GNSS signals are blocked, which leads to a rapid deterioration of the positioning accuracy of the integration system [1,2]. To solve this problem, researches on multi-sensor autonomous navigation are conducted all over the world.

However, high precision sensors cannot be widely equipped in a swarm because of size, weight, power and cost, which is summarized as a SWPAC problem. Therefore, CN began to pay greater attention to researchers in the robot and navigation field, with an interest in improving the positioning accuracy of other LANs by sharing the navigation information of HANs in the swarm. Advanced sensors such as depth camera, lidar and ultra-wide band (UWB) are applied to observe the relative position, attitude or velocity between nodes in swarm.

According to mathematical principles, the constraints in the three-dimensional coordinate between nodes can be established by using the relative range and angle between nodes in the swarm, which makes it possible for the accurate positioning of LANs based on navigation information obtained by HANs.

At present, UWB is generally installed on various unmanned platforms to provide the relative range between nodes in the research of CN. In general, this small sensor generally weighs less than 0.2 kg and consumes less than 10 w. The ranging error of UWB could be less than 0.5 m when measuring the target within 100 m under ideal conditions. In the published papers, many scholars use the relative range and angle information between nodes to conduct sufficient CN simulation experiments on different unmanned platforms [3,4]. In their simulation, the observation in the filter is the state of nodes [4], which is not suitable for the IMU-based unmanned platform to carry out state prediction according to the next step of the Kalman filter. Paper [5] proves that although IMU-based LANs could improve their positioning accuracy by CN, the attitude estimation would be divergent because the motion equation of the IMU-based unmanned platform is highly nonlinear and the higher-order term is always neglected in the Taylor expansion. The error of attitude cannot be well compensated and affects the filter, leading to the deterioration of the CN performance of the whole swarm. To solve this problem, a new method was needed to obtain the relative attitude then calculate the optimal result of attitude estimation of LANs through filters using the output of IMU installed on the platform.

Much work has been done on collaborative navigation to improve positioning accuracy. However, the research focusing on attitude accuracy could be more satisfactory. On the one hand, high-precision attitude estimation is an important means of precise cooperative navigation but research literature rarely mentions which sensor can be used to directly obtain the angle between two nodes. On the other hand, accurate attitude determination is the basis for the cooperative swarm to carry out more work. Taking the target detection work as an example, just taking the position information of each node cannot calculate the coverage of airborne radar, camera, and other sensors of the node in real time, nor can it achieve the tracking of moving targets.

In a word, how to obtain the relative attitude with high accuracy has become an urgent problem for small robot cooperative navigation. In reference [6], the researchers used a complex UWB array to locate and orient the ground robot. The orientation accuracy is about 3 degrees when applying Kalman filtering, and the root mean square error (RMSE) is reduced to about 1.5 degrees after using the graph optimization algorithm based on the slide window. However, this method cannot apply to all unmanned swarms. First, complex UWB arrays will cost a great deal; Second, the UWB array layout requires a larger platform; Third, the severe electromagnetic signal interference will block communication between nodes in the swarm.

In order to design a method that is easy to realize in practice, in this paper we selected a camera commonly used by UAV to solve this problem. According to the principle of polar geometry, the 3D coordinates of the target in the camera frame (Frame-C) can be calculated by the coordinate in pixel frame (Frame-P) of the target and the distance from the lens. When three or more identifiable targets are in the overlap of a HAN and a LAN, the corresponding vectors can be obtained by connecting the targets, which should be more than three as well. For convenience, the vectors obtained by camera are called visual vector in this paper. The Euler between different nodes can be calculated by algorithms for multivector attitude determination using visual vectors' projection in Frame-C.

Attitude determination is a measurement technology based on relative positioning [7]. By observing two or more baselines at the same time, the three-dimensional attitude calculation of the carrier can be realized, which is widely used in aerial, marine and land navigation. Generally, there are two methods to determine the three-axis attitude:

- Deterministic algorithm, such as TRIAD (Tri-Axial Attitude Determination) [8].
- Optimization algorithm, such as QUEST (Quaternion Estimator) [9] or SVD (Singular Value Decomposition) [10].

When using a double baseline to solve an attitude determination problem, deterministic algorithm and several optimal algorithms could all work, but different algorithms have their own advantages and disadvantages in various application scenarios.

SVD (1968) was first proposed by Markley, but, it was not widely used in practice in the 1960s–1980s because of the large amount of calculation when the computational power was seriously limited. In this century, with the rapid development of computer computing power, Wahba's problem has also been applied in more fields. Yang used the general root of quartic equation to solve the optimal quaternion in 2013. The algorithm is fast, but a problem still exists. It may not have a real root of characteristic polynomial, which will lead to plural quaternion [11]. So in 2015, Yang introduced the idea of Riemannian manifold and developed a more robust iterative method [12]. In 2017, Wu Jin designed a new fast linear attitude estimator (FLAE) [13] where the analytical method and iterative method for solving the eigenvalues are provided at the same time.

Multi-vector attitude determination algorithm has been studied to solve the relative attitude problem between vehicles. In fact, the early research of attitude determination algorithm mainly comes from solving the problem of satellite attitude determination. The attitude of the body in the inertial coordinate system can be obtained by calculating the installation position of the star sensor in the body.

For solving the problem of space non-cooperative target relative attitude determination, Wang proposed a relative attitude determination method of target spacecraft using dual-feature structures [14]. Zhang Lei [15] introduced the deduction that the attitude measurement accuracy is not only affected by the relative position between the navigation stars described by the condition number, but also related to the different positions of the navigation star group in the star map.

Nevertheless, the application environment attitude determination faced in multi-unmanned platform CN swarm is different from traditional applications such as satellite attitude determination, which is mainly perceived in the following three aspects:

The first is much larger measurement noise. In the application of satellite attitude determination, signal-to-noise ratio (SNR) is about 2000 db which is an inaccessible precision for a camera with ranging functions such as a depth camera or a binocular camera installed on unmanned platforms.

The second is false-alarm picture in practice. We adopt a YOLO-deep-sort algorithm based on YOLOv5 edition6 to detect targets. Although this algorithm is very friendly to the low computational power platform because of its great performance on lightweight, the occasional target misjudgment is fatal to the traditional algorithm.

The third is the higher demand of rapidity and stability. As mentioned above, extracting the image features of the target is a relatively complex process for most low-cost unmanned platforms, and high-definition image of the target is necessary as well during this process. If the default observation caused by target invisibility or overtime calculation occurs in the process of navigation and attitude determination for a long time, the divergence of results will be irreversible.

As mentioned above, divergence of attitude always occurs when the extended Kalman Filter (EKF) or other linear filtering methods are directly used. At the same time, the uncertainty of detection is also not suitable for directly using the calculated relative angle as the observation of the filter. More significantly, the biggest difference between CN and single node positioning is that multiple UUVs can coordinate, cooperate and share their own state with other nodes in a CN swarm [16], which means that every node needs to continuously send or receive information from each platform in the swarm. All status related to observational status of moment parameters must be changed when updating observables, which will increase the calculations and make communication difficult. Thus it will hinder the application of CN when using the Kalman filter to fuse multi-sensors' data. As described in [17] and [18], it can be seen from simultaneous localization and robot mapping that the information filter (IF) performs better than a Kalman filter. Therefore,

augmented IF with Euler as status and relative angle as observation is presented to use in CN to determine the attitude of nodes in the swarm.

The efficiency of cooperative navigation is mainly reflected in the fusion of the observation of all nodes in the swarm. In a decentralized swarm, each node needs to process its own sensing information and fuse it with those from other nodes to calculate the position and posture, which means that each node needs to process a large amount of information. The difficulty caused by time synchronization makes data fusion more complex. Different from the traditional Kalman Filter, the IF uses information parameters in its iterative optimization so that it only needs to update the parameters related to the observation without updating the status of all nodes in the swarm. The observation update of information filtering is local, so the computational power and communication consumption will be less, which makes it very suitable for cooperative navigation swarms that need to fuse a large amount of nonsequential information [19].

Considering the practical problems mentioned above, we propose a new method to determine the attitude of unmanned platforms by visual vectors. Based on CN IF, this method obtains the relative-attitude Euler angle of nodes in the swarm quickly and accurately as the observation to calculate the attitude of LANs. The key points of our work are as follows:

- Fast and robust attitude determination algorithm by visual vectors
- Robust IF construction for IMU-based vehicle

The remainder of this paper is organized as follows. Section 2 describes the details of the attitude determination of unmanned platforms in CN swarm algorithm. Then in Section 3 we continue with the simulation of multi-vector attitude determination algorithm and attitude determination based on CN to verify the performance of the proposed method. In Section 4, the analysis and discussion of the experimental results are presented to evaluate the performance of the algorithm. Section 5 discusses conclusions and future work.

## 2. Materials and Methods

As shown in Figure 1, attitude determination of unmanned platforms in the CN swarm mainly needs to solve three problems:

- Target detection and calculation for coordinates in the navigation frame (Frame-N).
- Determination of the relative angle between nodes in the swarm.
- Attitude error compensation based on decentralized CN IF.

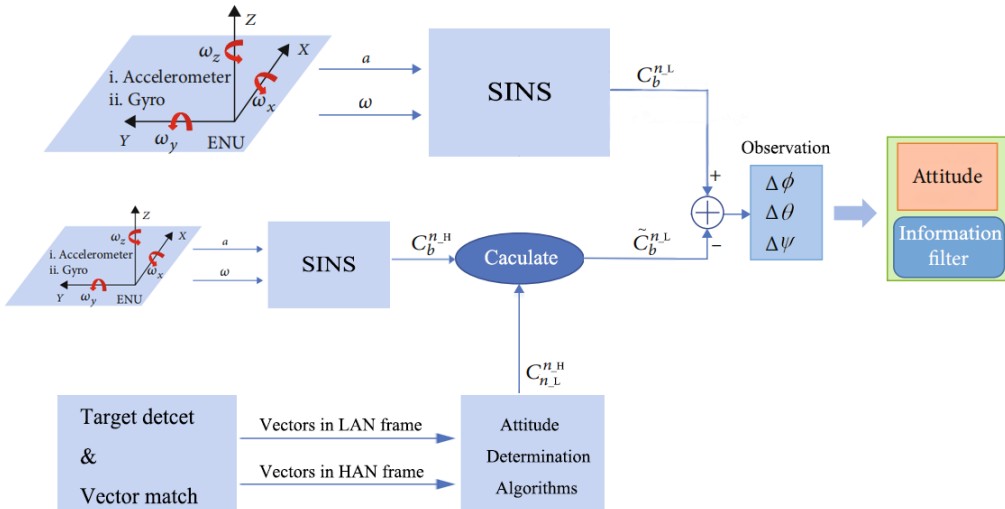

**Figure 1.** Method to compensate for the error of attitude for LANs in CN swarm.

Among the three points mentioned, designing an improved algorithm for calculating the attitude-transfer matrix using visual vectors and compensating for the error of attitude

angle of LANs based on IF constitute the main emphasis and innovation point of this paper. In Section 2.1, the approach of constructing vectors from detected targets is described in detail. Then the fast and robust multi-vector attitude determination algorithm is derived in Section 2.2 and IF is constructed in Section 2.3.

### 2.1. Frames Involved and Method to Build Visual Vectors

In this paper, the installed cameras are fixed on the platforms, therefore, Frame-C can convert to body frame (Frame-B) through a known and invariant attitude conversion matrix. Consequently, we can calculate the attitude conversion matrix $C_{b1}^{b2}$ between platforms as long as we calculate the relative attitude relation between the cameras.

Related formula are proposed as Equation (1)

$$\begin{bmatrix} \psi_{b1} \\ \theta_{b1} \\ \gamma_{b1} \end{bmatrix} = C_{c1}^{b1} \begin{bmatrix} \psi_{c1} \\ \theta_{c1} \\ \gamma_{c1} \end{bmatrix}, \begin{bmatrix} \psi_{b2} \\ \theta_{b2} \\ \gamma_{b2} \end{bmatrix} = C_{c2}^{b2} \begin{bmatrix} \psi_{c2} \\ \theta_{c2} \\ \gamma_{c2} \end{bmatrix}, \quad C_{b1}^{b2} = C_{b1}^{c1} \cdot C_{c1}^{c2} \cdot C_{c2}^{b2} \tag{1}$$

where $C_{b1}^{c1}$ and $C_{b2}^{c2}$ are the attitude conversion matrix which reflects the installation relationship of the camera. They should be invariant in practice so the key to calculate the relative angle is to calculate $C_{c1}^{c2}$.

According to the requirements of multivector attitude determination algorithm, the projection of at least two different vectors in two frames is required to obtain the attitude transformation matrix between these two frames. As shown in Figure 1, the first step to construct the vector in Frame-C is to detect targets and get their coordinates. Next, connect center points of the box of the target in the proper order. If more than two sets of coordinates are correctly obtained in the first step, we can get enough visual vectors to determine the relative attitude between two platforms.

Besides the targets we need to detect and recognize, the cooperative nodes in the overlap horizon can also help to construct vectors. In general, it should not be hard to find three or more targets in the overlap to construct more than three sets of visual vectors.

As shown in Figure 2, we do not discuss exceptional cases in this paper because how to conduct target recognition is not the focus of our research. In our experiment, we apply pruning YOLOv5, which could work at 20 Hz on our experimental platform to detect targets efficiently.

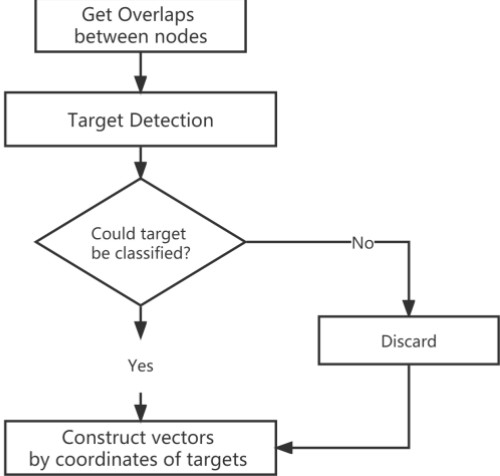

**Figure 2.** Method to construct a visual vector.

Figure 3 shows how to construct visual vectors in detail. The mathematical description is given as follows: Define the sets of detected targets for platform i and platform j as $\Omega_i = \{o_1^i, o_2^i, o_3^i \ldots\}$ and $\Omega_j = \{o_1^j, o_2^j, o_3^j \ldots\}$. Take the union of these two sets as $\Omega_{ij} = \Omega_i \cap \Omega_j$, which represents the targets in overlap. Only when $card(\Omega_{ij}) \geq 3$ can the

attitude determination algorithm work. *card*(·) represents the operation of calculating the number of elements in a set.

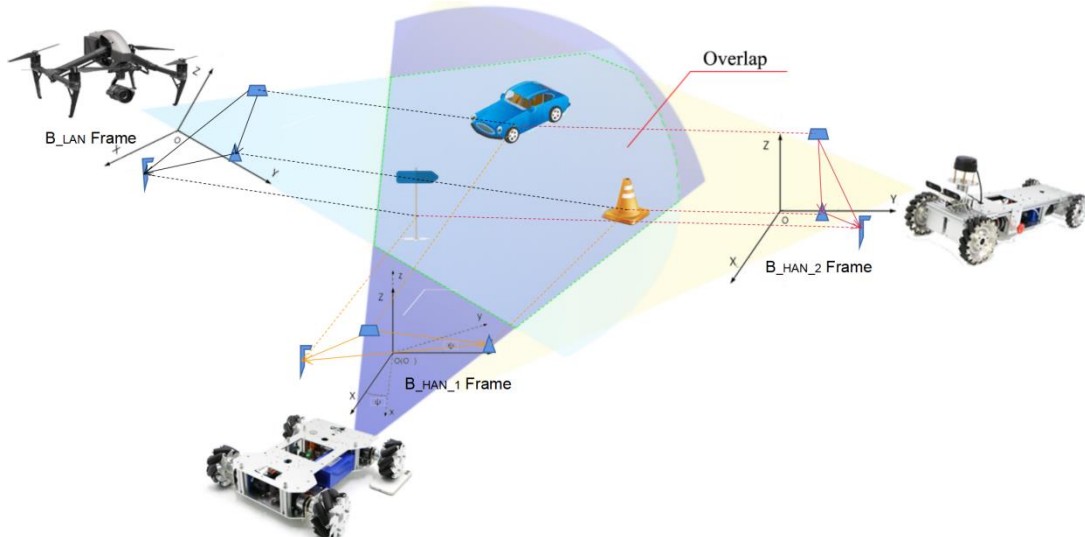

**Figure 3.** Construct visual vectors by targets in the overlap (covisibility graph).

When it comes to the situation that more than one intra-class target are detected, the problem becomes much more complex. YOLO cannot identify intra-class targets, so different platforms might have different identifications of the same target or misjudge two different targets as the same one [20]. If the cooperative nodes are detected in overlap, we might identify it by communication but this method cannot work when facing other targets. Employing JPDA (Joint Probabilistic Data Association) or Dead-Reckoning to identify unclassified targets by their different trajectory could solve this problem but it requires greater computation at the same time. We are not investigating this complex case in this paper. In the simulation in Section 4 and the experience in Section 5, we also place different kinds of objects in the experimental site.

### 2.2. Improved QUEST for Multi-Vector Attitude Determination

Because of its optimum balance of stability, rapidity and accuracy, QUEST has become the most prevalent attitude determination algorithm. Considering that the vector accuracy obtained by the unmanned platform attitude measurement is much lower, we need to make some improvements to enhance its robustness in the case of insufficient vector accuracy.

QUEST is also a algorithm based on Wahba's problem. Wahba's problem is to convert the direction measurement into attitude measurement. Its specific expression is as follows: if there are $n$ unit vectors in the $Frame - B$ which are recorded as $v_i$, $i = 1, \ldots n$. The value obtained by measuring these unit vectors in another $Frame - B'$ is $\hat{v}_i$, $i = 1, \ldots n$. The problem is to find a rotation matrix $C$ to minimize the loss function.

$$\mathrm{L(C)} = \frac{1}{2} \sum_{i=1}^{n} \mathrm{w_i} \| v_i - C\hat{v}_i \|^2 \tag{2}$$

where $\mathrm{w_i}$ represents the weight factors for each obviation vector. $\mathrm{w_i}$, $i = 1, \ldots n$, are a set of positive weights satisfying $\sum_{i=1}^{n} \mathrm{w_i} = 1$, usually chosen as $\mathrm{w_i} = 1/\sigma_i^2$, with $\sigma_i^2$ the variance parameters of the measurement vectors.

To meet the requirements of the application, the use of attitude rotation quaternions can effectively reduce the amount of computation required in the solution process by reducing the number of unknowns. The loss function is expressed in the form of quaternion as Equation (3):

$$\mathrm{tr}(BC^{\mathrm{T}}) = \mathrm{q}^{\mathrm{T}} K \mathrm{q} \tag{3}$$

where

$$K = \begin{bmatrix} S - I_{3X3}tr(B) & z \\ z^T & tr(B) \end{bmatrix} \quad z = \sum_{i=1}^{n} a_i b_i \times r_i \quad S = B + B^T \tag{4}$$

Utilizing Lagrange operator to calculate the maximum value of Equation (4), we get:

$$K q_{opt} = \lambda \vec{q}_{opt} \tag{5}$$

If and only if $\lambda$ is equal to the maximum eigenvalue of matrix $K$, the eigenvector of $\lambda_{max}$ is the optimal quaternion $q_{opt}$. Therefore, we can rearrange Equation (5) as

$$[(\lambda_{max} + tr(B))I - S]\vec{q} = q_4 z \tag{6}$$

$$(\lambda_{max} - tr(B))q_4 = \vec{q}^T z \tag{7}$$

From Equations (6) and (7), the inverse of the matrix is expressed by the adjoint matrix, then we get

$$\begin{aligned} \vec{q} &= q_4[(\lambda_{max} + tr(B))I - S]^{-1}z \\ &= \{q_4/\det[(\lambda_{max} + tr(B))I - S]\}adj[(\lambda_{max} + tr(B))I - S]z \end{aligned} \tag{8}$$

Therefore, we describe $\vec{q}$ in terms of Gibbs vector and normalize it then we get:

$$\vec{q} = \frac{1}{\sqrt{1^2 + |Y|^2}} \begin{bmatrix} Y \\ 1 \end{bmatrix} \tag{9}$$

where

$$Y = [(\lambda + \sigma)I - S]^{-1}Z \tag{10}$$

$$\lambda = \sigma + Z^T \cdot Y \tag{11}$$

Inserting Equation (11) into Equation (9) would mean that the $\lambda$ meets the $q_{opt}$ we need as:

$$\lambda = \sigma + Z^T[(\lambda + \sigma)I - S]^{-1}Z \tag{12}$$

Equation (12) is equivalent to the characteristic equation for the eigenvalues of $K$. To avoid the problems posed by this singularity with the consideration on the problem that the Gibbs vector becomes infinite when the angle of rotation is $\pi$, we derive an expression that permits the computation of $\vec{q}_{opt}$ without the intermediary of the Gibbs vector. For an eigenvalue $\xi$ of any square matrix $S$ satisfies the characteristic equation:

$$\det|S - \xi I| = 0 \tag{13}$$

$$-\xi^3 + 2\sigma\xi^2 - \kappa\xi + \Delta = 0 \tag{14}$$

According to the Cayley-Hamilton theorem [21], $S$ should satisfy the same equation. Then we get a convenient expression for the characteristic equation:

$$\lambda^4 - (a + b)\lambda^2 - c\lambda + (ab + c\sigma - d) = 0 \tag{15}$$

where

$$\left\{ \begin{array}{l} \sigma = tr(B) \\ \kappa = tr(adj(S)) \\ \Delta = \det(S) \end{array} \right. \left\{ \begin{array}{l} a = \sigma^2 - \kappa \\ b = \sigma^2 + z^T z \\ c = \Delta + z^T S z \\ d = z^T S^2 z \end{array} \right. \tag{16}$$

Since $\mathrm{tr}(\boldsymbol{B}\boldsymbol{C}^{\mathrm{T}})$ is a small quantity causing $\lambda$ is very close to 1, its initial value can be set to $\lambda_0 = 1$ [22]. Then, the Newton iteration can be conducted using:

$$\lambda(n+1) = \lambda(n) - \frac{f[\lambda(n)]}{f'[\lambda(n)]} \tag{17}$$

Usually, $\lambda$ can be very accurate after several iterations in traditional QUEST algorithms. When the eigenvalue is obtained, the eigenvector can then be calculated using elementary row operations. However, as the accuracy is not linear with iteration times, fixed iteration times will not always achieve good results. Different from the method proposed by Markley, a novel symbolic approach is investigated for increasing the speed of calculating. For this purpose, Equation (15) can then be calculated as follows:

$$
\begin{aligned}
\lambda_1 &= \frac{1}{2\sqrt{6}}\left(\Gamma_2 - \sqrt{-\Gamma_2^2 + 12(a+b) + \frac{12\sqrt{6}\cdot c}{\Gamma_2}}\right) \\
\lambda_2 &= \frac{1}{2\sqrt{6}}\left(\Gamma_2 + \sqrt{-\Gamma_2^2 + 12(a+b) + \frac{12\sqrt{6}\cdot c}{\Gamma_2}}\right) \\
\lambda_3 &= -\frac{1}{2\sqrt{6}}\left(\Gamma_2 - \sqrt{-\Gamma_2^2 + 12(a+b) + \frac{12\sqrt{6}\cdot c}{\Gamma_2}}\right) \\
\lambda_4 &= -\frac{1}{2\sqrt{6}}\left(\Gamma_2 + \sqrt{-\Gamma_2^2 + 12(a+b) + \frac{12\sqrt{6}\cdot c}{\Gamma_2}}\right)
\end{aligned}
\tag{18}
$$

With the parameters

$$
\begin{aligned}
\Gamma_0 &= -2(a+b)^3 + 27c_2^2 + 72(a+b)(ab + c\sigma - d) \\
\Gamma_1 &= \left(\Gamma_0 + \sqrt{-4\left((a+b)^2 + 12(ab + c\sigma - d)\right)^3 + \Gamma_0^2}\right)^{\frac{1}{3}} \\
\Gamma_2 &= \sqrt{4(a+b) + \frac{2^{\frac{3}{4}}\left((a+b)^2 + 12(ab + c\sigma - d)\right)}{\Gamma_1} + 2^{\frac{2}{3}}\Gamma_1}
\end{aligned}
\tag{19}
$$

Then, $\lambda$ is chosen by the value that is nearest to one. In this way, the solving process of $\lambda$ is significantly shortened. The use of a general solution makes traditional QUEST a faster attitude solution algorithm with less floating-point operation. Inserting the $\lambda_{\max}$ calculated by Equation (18) into Equation (9), we obtain the optimal attitude quaternion $\vec{q}_{\mathrm{opt}}$ as follow:

$$
\begin{aligned}
\vec{q}_{\mathrm{opt}} &= \frac{1}{\sqrt{\gamma^2 + |x|^2}}\begin{bmatrix} x \\ \gamma \end{bmatrix} \\
&= \frac{\begin{bmatrix} \alpha\boldsymbol{I} + (\lambda_{\max} + \mathrm{tr}(\boldsymbol{B})) + \boldsymbol{S}^2 \\ \alpha(\lambda_{\max} + \mathrm{tr}(\boldsymbol{B})) - \det(\boldsymbol{S}) \end{bmatrix}}{\sqrt{[\alpha(\lambda_{\max} + \mathrm{tr}(\boldsymbol{B})) - \det(\boldsymbol{S})]^2 + \left|\left(\alpha\boldsymbol{I} + (\lambda_{\max} + \mathrm{tr}(\boldsymbol{B})) + \boldsymbol{S}^2\right)\right|^2}}
\end{aligned}
\tag{20}
$$

### 2.3. Attitude Angle Determination Based on IF

Cooperative navigation technology is widely used in unmanned platforms. The common method is data fusion with improved Kalman filters based on specific navigation sensors. As shown in [15], when platforms are defined as a processing node, the Kalman filter can fuse navigation information of each node to the master node in a centralized Kalman filter algorithm.

Assume that the unmanned swarm consists of $n$ nodes, the motion model and observation model for an IMU-based platform of attitude compensation in cooperative navigation are:

$$X_{k+1} = \begin{bmatrix} f^1(x_k^1, u_k^1) \\ f^2(x_k^2, u_k^2) \\ \vdots \\ f^N(x_k^N, u_k^N) \end{bmatrix} + W_k \triangleq \boldsymbol{\Phi}(X_k, U_k) + W_k \tag{21}$$

where, $X_{k+1}$ and $X_k$ represent the status of the whole CN swarm at the time $k+1$ and $k$. Furthermore, $x_k^i$ represents the status of platform-i at the time $k$ and we have $X_k = \begin{bmatrix} x_k^1 & x_k^2 & \dots & x_k^N \end{bmatrix}$.

$f^i$ is status transition matrix of platform-i, which is determined by the dynamics of this platform. $u_k{}^i$ is the parameter required for state estimation, $w_k{}^i$ is the system noise of platform-i at the time $k$, and the variance is $Q_k{}^i$, meets $w_k{}^i \sim N(0, Q_k^i)$ and $W_k = \begin{bmatrix} w_k^1 & w_k^2 & \dots & w_k^N \end{bmatrix}$. The single-platform observation update equation at time k and the inter-platform observation equation of platform-i to platform-j can be expressed as:

$$\begin{aligned} z_k^i &= h^i\left(x_k^i\right) + v_k^i \\ z_k^{ij} &= h^{ij}\left(x_k^i, x_k^j\right) + v_k^{ij} \end{aligned} \tag{22}$$

where $z_k^i$ and $z_k^{ij}$ represents the single platform observation from platform-i to itself and platform-j. $h^i$ and $h^{ij}$ is the observation equation, which should suit the observation in the filter and the sensors used. $v^i$ and $v^{ij}$ is the observation noise and follows $v_k{}^i \sim N(0, R_k^i)$ and $v_k{}^{ij} \sim N(0, R_k^{ij})$ respectively.

1.    Traditional decentralized CN Kalman filtering

For an unmanned swarm, the observation update of the single platform at the time k is

$$\hat{X}_{k+} = \begin{bmatrix} \hat{x}_{k-}^1 \\ \hat{x}_{k-}^2 \\ \vdots \\ \hat{x}_{k-}^N \end{bmatrix} + \begin{bmatrix} \Sigma_{\hat{X}_{k+}}^{1i} \\ \Sigma_{\hat{X}_{k+}}^{2i} \\ \vdots \\ \Sigma_{\hat{X}_{k+}}^{Ni} \end{bmatrix} (\nabla h^i)^T (s^i)^{-1} v^i \tag{23}$$

$$\Sigma_{\hat{X}_{k+}} = \Sigma_{\hat{X}_{k-}} - \begin{bmatrix} \Sigma_{\hat{X}_{k-}}^{1i} \\ \Sigma_{\hat{X}_{k-}}^{2i} \\ \vdots \\ \Sigma_{\hat{X}_{k-}}^{Ni} \end{bmatrix} (\nabla h^i)^T (s^i)^{-1} \nabla h^i \begin{bmatrix} \Sigma_{\hat{X}_{k-}}^{i1} \\ \Sigma_{\hat{X}_{k-}}^{i2} \\ \vdots \\ \Sigma_{\hat{X}_{k-}}^{iN} \end{bmatrix}^T \tag{24}$$

At the initial time, the state parameters of nodes are not correlated. After their collaborative observation, the state parameters of nodes in the swarm are correlated. It can be seen from Equations (23) and (24) that even if the single platform only updates the observation involved with itself, all status of overall nodes in the unmanned swarm still needs to be updated according to the traditional Kalman filtering step. At the time k, platform-i updates its observation to platform-j as:

$$\hat{X}_{k+} = \begin{bmatrix} \hat{x}_{k-}^1 \\ \hat{x}_{k-}^2 \\ \vdots \\ \hat{x}_{k-}^N \end{bmatrix} + \begin{bmatrix} \Sigma_{\hat{X}_{k-}}^{1i} & \Sigma_{\hat{X}_{k-}}^{1j} \\ \Sigma_{\hat{X}_{k-}}^{2i} & \Sigma_{\hat{X}_{k-}}^{2j} \\ \vdots & \vdots \\ \Sigma_{\hat{X}_{k-}}^{Ni} & \Sigma_{\hat{X}_{k-}}^{1i} \end{bmatrix} \begin{bmatrix} (\nabla h_i^{ij})^T \\ (\nabla h_j^{ij})^T \end{bmatrix} (s^{ij})^{-1} v^{ij} \tag{25}$$

$$\Sigma_{\hat{X}_{k+}} = \Sigma_{\hat{X}_{k-}} - \Sigma_{\hat{X}_{k-}}^{Ni} \begin{bmatrix} \left(\nabla h_i^{ij}\right)^T \\ \left(\nabla h_j^{ij}\right)^T \end{bmatrix} \left(s^{ij}\right)^{-1} \begin{bmatrix} \left(\nabla h_i^{ij}\right)^T \\ \left(\nabla h_j^{ij}\right)^T \end{bmatrix} \Sigma_{ij}$$

$$\Sigma_{ij} = \begin{bmatrix} \Sigma_{\hat{X}_{k-}}^{i1} & \Sigma_{\hat{X}_{k-}}^{i2} & \cdots & \Sigma_{\hat{X}_{k-}}^{iN} \\ \Sigma_{\hat{X}_{k-}}^{1j} & \Sigma_{\hat{X}_{k-}}^{2j} & \cdots & \Sigma_{\hat{X}_{k-}}^{Nj} \end{bmatrix}$$

(26)

Similarly, we can see from Equations (25) and (26) that although only two platforms have been involved with the observation, the observation update needs to update all the platforms' statuses because the state parameters used in the CN Kalman filter are relevant.

Traditional decentralized CN Kalman filtering requires all platforms in the swarm to maintain time synchronization and continuously send or receive information from each other, which leads to a huge real-time communication burden. In addition, the observation update can only be carried out in sequence. For the observation at the same time, the swarm can only start the update of another observation after completing the update of the previous observation. As the number of the nodes in the swarm increases, the number of observations at the same time rises. This is why the traditional decentralized CN Kalman filtering is difficult to achieve in practice.

2.　　Decentralized CN Information Filter

IF uses information parameters to augment states and update observations, then recovers the status and covariance. This method preserves the historical states in the filter, thus the joint distribution of the information matrix is sparse and computational complexity of filter is less.

The Gaussian filter is widely used in estimated algorithms. It can be described by moment parameters (mean and variance $\{\hat{X}, P\}$) and information parameters (information vector and information matrix $\{\hat{y}, Y\}$). The relation between them is described as [23]:

$$\hat{y} = P^{-1}\hat{X}$$
$$Y = P^{-1}$$

(27)

The Kalman filter is a Gaussian filter based on moment parameters and IF is a Gaussian filter based on information parameters; thus they are equal in style. Their calculation features are as different as their different parameters. But their filtering steps are the same. Transforming the Kalman filtering of single-platform observation update at time k to the IF form, we get:

$$Y_{k+}^{-1} = Y_{k-}^{-1} - Y_{k-}^{-1} H_{k-}^T \left( H_{k-} Y_{k-}^{-1} H_{k-}^T + \left(P_k^i\right)^{-1} \right)^{-1} H_{k-} Y_{k-}^{-1}$$

(28)

where, $Y_{k-}$ and $Y_{k+}$ represents the prediction information matrix and observation update information matrix of nodes at time k. $H_{k-}^i$ denotes the observable linear matrix of platform-i. $P_k^i$ is the weight of observation value of platform-i. Inverse the left and right sides of the Equation (28) and the observation update in IF form shows as Equation (29):

$$Y_{k+} = Y_{k-} + H_{k-}^T (P_k) H_{k-} = Y_{k-} + \begin{bmatrix} 0 & \cdots & 0 & \cdots & 0 \\ \vdots & \ddots & \vdots & \ddots & \vdots \\ 0 & \cdots & I_k(i,i) & \cdots & 0 \\ \vdots & \ddots & \vdots & \ddots & \vdots \\ 0 & \cdots & 0 & \cdots & 0 \end{bmatrix}$$

(29)

From Equation (29), it is obvious that IF only needs to update the information matrix of platform-i instead of updating the whole platforms' status in the traditional way when updating the single platform observation. Similarly, as Equation (30) shows, it is only

necessary to update the information matrix related to platform-i and platform-i when updating the observation between this two platforms.

$$Y_{K+} = Y_{K-} + \left(H_{K-}^{ij}\right)^T P_K H_{K-}^{ij}$$

$$= Y_{K-} + \begin{bmatrix} 0 & \cdots & 0 & \cdots & 0 & \cdots & 0 \\ \vdots & \ddots & \vdots & & \vdots & \ddots & \vdots \\ 0 & \cdots & I_k(i,i) & \cdots & I_k(i,j) & \cdots & 0 \\ \vdots & & \vdots & \ddots & \vdots & & \vdots \\ 0 & \cdots & I_k(j,i) & \cdots & I_k(j,j) & \cdots & 0 \\ \vdots & \ddots & \vdots & & \vdots & \ddots & \vdots \\ 0 & \cdots & 0 & \cdots & 0 & \cdots & 0 \end{bmatrix}$$

(30)

$$\left.\begin{aligned} I_k(i,i) &= \left(\nabla h_i^{ij}\right)^T P_k^{ij} \nabla h_i^{ij} \\ I_k(i,j) &= \left(\nabla h_i^{ij}\right)^T P_k^{ij} \nabla h_i^{ij} \\ I_k(j,i) &= \left(\nabla h_j^{ij}\right)^T P_k^{ij} \nabla h_j^{ij} \\ I_k(j,j) &= \left(\nabla h_j^{ij}\right)^T P_k^{ij} \nabla h_j^{ij} \end{aligned}\right\}$$

3.　Design of state and observation in IF

This paper selects the following parameters as state variables: attitude angle error $\delta\varphi$, horizontal velocity error $\delta V$, zero offset of accelerometer $\varepsilon b$, gyroscope drift $\nabla b$.

For attitude compensation, the system state vector $X$ of IMU-based autonomous integrated navigation system can be defined as

$$X = \begin{bmatrix} \delta\phi & \delta\theta & \delta\psi & \delta v_N & \delta v_E \end{bmatrix}$$ (31)

This is because the ground (or sky) directional of the inertial navigation system is divergent and generally cannot be solved without external reference information or damping, so the ground velocity error state and height error state should be removed. The position error is that the velocity error is related as well. When there is no separate position reference information, the position error state should be removed.

Inertial Navigation System Error State as:

$$\dot{X} = F \cdot X + G \cdot u$$ (32)

where $F$ is the system state matrix and $G$ is the systematic error matrix.

$$u = \begin{bmatrix} \delta w_{ib,x}^b & \delta w_{ib,y}^b & \delta w_{ib,z}^b & \delta f_x^b & \delta f_y^b \end{bmatrix}$$

$$F = \begin{bmatrix} 0 & -\Omega \sin L - \frac{v_E \tan L}{R_E} & \frac{v_N}{R_N} & 0 & \frac{1}{R_E} \\ \Omega \sin L + \frac{v_E \tan L}{R_E} & 0 & \Omega \cos L + \frac{v_E}{R_E} & \frac{-1}{R_N} & 0 \\ \frac{-v_N}{R_N} & -\Omega \cos L - \frac{v_E}{R_E} & 0 & 0 & -\frac{\tan L}{R_E} \\ 0 & -f_D & f_E & \frac{v_D}{R_E} & -2\Omega \sin L - \frac{2 v_E \tan L}{R_E} \\ f_D & 0 & -f_N & 2\Omega \sin L + \frac{v_E \tan L}{R_E} & \frac{v_N \tan L}{R_E} + \frac{v_D}{R_E} \end{bmatrix}$$ (33)

The selection of the measurement vector of the integrated navigation system is directly related to the measurement accuracy of the entire system. Based on the known matching results, the attitude observation equations are established.

In the loose combination observation method, the observation is the relative angle obtained by the sensor. Since the error state is selected to establish the state equation in

the filter, the form of the observation equation will be very simple without considering the lever arm effect.

Set the observation as Equation (34)

$$Z = \begin{bmatrix} \delta\phi & \delta\theta & \delta\psi \end{bmatrix}$$
$$= \begin{bmatrix} \phi_{ob} - \phi & \theta_{ob} - \theta & \psi_{ob} - \psi \end{bmatrix} \tag{34}$$

We have $H = I_{3\times3}$. The error of attitude angle in three axes is selected as observation, in this way, the observation matrix H will be linear. The observation is directly obtained by subtracting the angle measured by the sensor from the attitude angle calculated by the IMU inertial navigation solution. Although the accuracy of loose combination will decrease compared with that of tight combination, the linear observation equation of loose combination will not introduce high-order error.

## 3. Simulation

### 3.1. Simulation of Improved Algorithm for Multi-Vector Attitude Determination

This section evaluates the performance of the proposed algorithm via the Monte Carlo simulation.

The mathematical equivalence of various attitude algorithms can be derived strictly. Strict mathematical derivation is complex and difficult. This section verifies the above algorithms from the perspective of simulation. The verification vector uses cases 1–6 set by Markley [24].

Each test case is specified by a set of measurement vectors and measurement noise. These cases model different application scenarios, such as three fine sensors with orthogonal boresight, one fine and one coarse sensor and measurements in a sensor with a small field of view. The robustness and accuracy of attitude determination algorithms could be evaluated effectively and comprehensively.

We set C_truth = [0.352, 0.864, 0.360; −0.864, 0.152, 0.480; 0.360, −0.480, 0.800]. Then calculate the loss function and misalignment angle error of QUEST, SVD, ESOQ, FLAE, FOAM and Davenport algorithms respectively and count the time taken by running it 2000 times. The computer running simulation is equipped with Intel Core i5-8400 CPU. The results are shown in Tables 1–3. In these three tables, the part filled with blue means it performs better than the new algorithm we proposed. Similarly, green and red parts represent equal and worse performance respectively.

**Table 1.** Estimation Error. The part filled with blue means it performs better than the new algorithm we proposed. Similarly, green and red parts represent equal and worse performance respectively.

| Case | Davenport | FLAE | QUEST | Improved QUEST |
|------|-----------|-------|-------|----------------|
| 1 | 0.517 | 0.523 | 0.522 | 0.518 |
| 2 | 0.616 | 0.615 | 0.616 | 0.616 |
| 3 | 0.518 | 0.521 | 0.523 | 0.516 |
| 4 | 0.616 | 0.617 | 0.616 | 0.615 |
| 5 | 0.617 | 0.616 | 0.617 | 0.617 |
| 6 | 0.574 | 0.574 | 0.573 | 0.572 |

**Table 2.** Loss Function. The meanings of different colors are the same as Table 1.

| Case | Davenport | FLAE | QUEST | Improved QUEST |
|------|-----------|-------|-------|----------------|
| 1 | 0.09 | 0.147 | 0.154 | 0.09 |
| 2 | 0.102 | 0.104 | 0.103 | 0.101 |
| 3 | 0.09 | 0.136 | 0.154 | 0.09 |
| 4 | 0.101 | 0.102 | 0.102 | 0.103 |
| 5 | 0.104 | 0.103 | 0.101 | 0.102 |
| 6 | 0.900 | 0.959 | 0.958 | 0.91 |

**Table 3.** Time used (for 2000 times)/s. The meanings of different colors are the same as Table 1.

| Case | Davenport | FLAE | QUEST | Improved QUEST |
|------|-----------|------|-------|----------------|
| 1 | 0.501 | 0.285 | 0.329 | 0.366 |
| 2 | 0.326 | 0.22 | 0.232 | 0.295 |
| 3 | 0.399 | 0.22 | 0.286 | 0.296 |
| 4 | 0.309 | 0.20 | 0.254 | 0.263 |
| 5 | 0.299 | 0.24 | 0.251 | 0.252 |
| 6 | 0.330 | 0.22 | 0.286 | 0.280 |

During observation, we set the SNR of vector observation to 20 dB. From the above table we see that the algorithms with poor robustness such as FOAM and ESOQ find it difficult to solve the stable attitude when the vector observation is not accurate enough. Due to the improvement of CPU performance, there is little difference in computing speed between the algorithms in the case of fewer observation vectors. Compared with QUEST and FLAE, our improved algorithm does not perform well enough, especially when it comes to the cost of time, but its estimation error reflects the robustness dealing with imprecise observation is much lower than other algorithms.

### 3.2. Simulation of IF for Attitude Error Compensation

In Section 3.1, we established a new observation model and filter for the INS-Camera CN attitude determination system, and proposed a concrete method of implementation. In this section, we applied trajectory and IMU data generated by simulation to verify the effectiveness of the designed an IF for attitude compensation. In this simulation, the error can be considered to be composed of constant deviation and white noise.

The simulation data simulated the trajectory of the unmanned vehicle in the Frame-N and the time lasted 290 s. The trajectory is shown in Figure 4 and yaw of LAN is shown in Figure 5a.

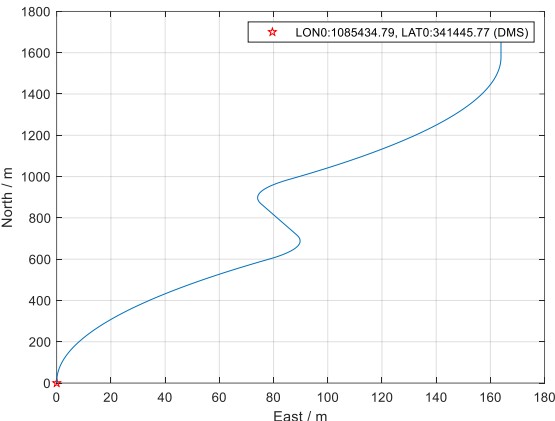

**Figure 4.** The trajectory of the unmanned vehicle in the Frame-N.

The yaw calculated after the filtering process is shown in Figure 5b. Compared with Figure 5a, it is obvious that the attitude divergence caused by the cumulative error of IMU has been solved. Figure 6 shows that the root mean square error (RMSE) of attitude angle in three axis is less than two degrees.

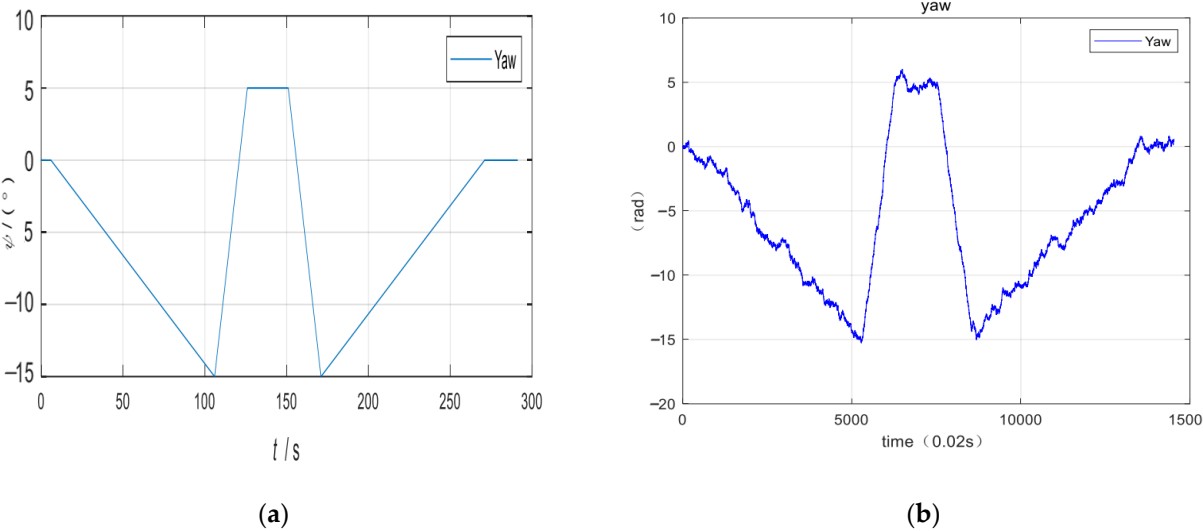

(**a**)  (**b**)

**Figure 5.** The yaw of the platform: (**a**) The true value of yaw; (**b**) The yaw of the platform after compensation.

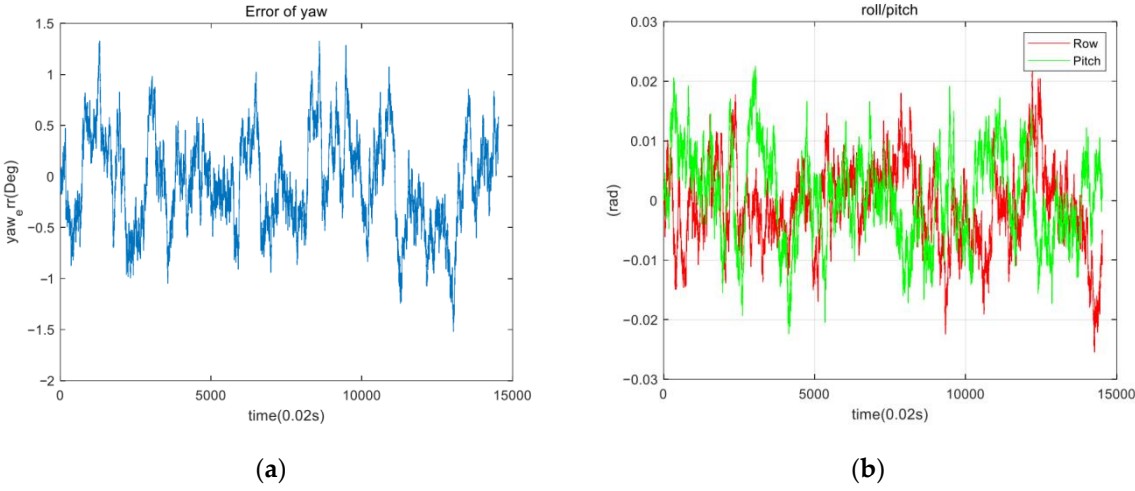

(**a**)  (**b**)

**Figure 6.** The error of attitude angle of the platform after compensation: (**a**) yaw; (**b**) roll and pitch.

## 4. Experimental Results and Discussion

### 4.1. Introduction of Platforms and Site Setting

In this paper, we select two wheeled robots and one UAV as experimental platforms. For convenience, we names them Node 1, Node 2, and Node 3. They are shown in Figure 7.

Among them, Node 1 act as a HAN that installs high-precision INS and Node 2 is a LAN that installs low-precision INS. Due to the lack of attitude reference, our UAV is not suitable to observe others. So we employ it just as a cooperative node for being observed by others in the swarm. The output of high precision INS are used as references of the attitude of Node 1.

To improve the success rate of target recognition, targets for detection should be highly visible and easily distinguishable. According to the above requirements, we set the position of targets and nodes as in Figure 8.

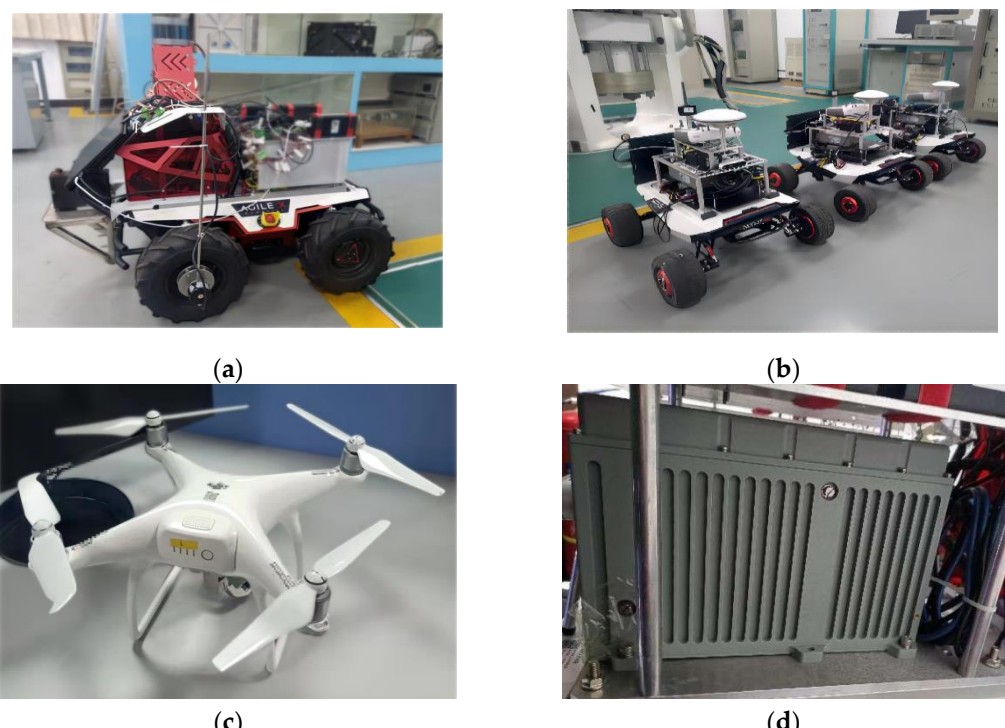

**Figure 7.** Experimental platforms and major sensors installed. (**a**) Node 1: HAN; (**b**) Node 2: LAN; (**c**) Node 3: drone; (**d**) High-precision INS.

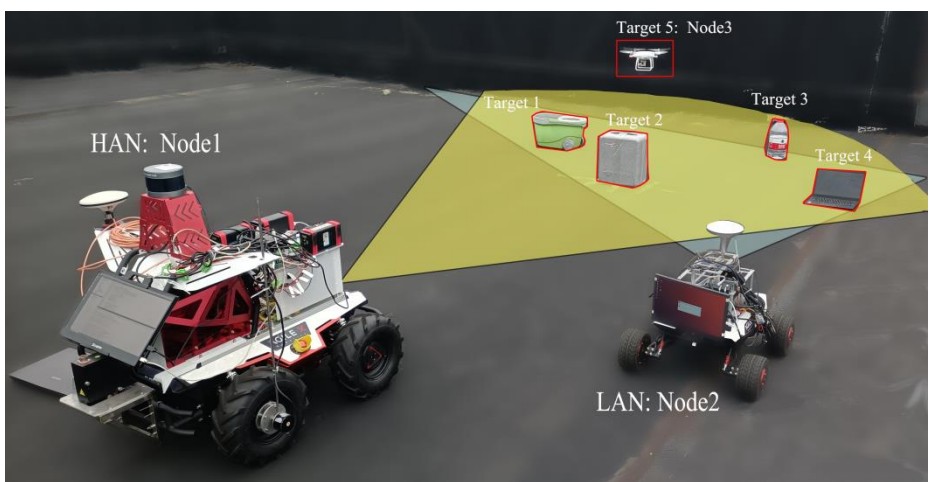

**Figure 8.** Full view of the experimental site.

### 4.2. Experiment Plan

However, the output of IMU on the UAV is not open source so we lost the attitude reference of this platform. For this reason, we just set it stay still to perform as an observed target.

One key point of the method we proposed is to obtain the relative attitude angle between different nodes. To ensure the covisibility graph could cover the entire experimental site, we let Node 1 with high-precision INS rotate in place, ensuring that the five targets can be observed. At the same time, Node 2 and Node 3 in the Figure 8 would stay still in their initial position. The whole movement lasts for 18 s.

In this process, we can calculate the coordinates of targets as shown in Figure 9 and obtain the visual vectors in Frame-C of different nodes. Using the different sets of

visual vectors, we can calculate the relative attitude between each nodes based on the improved QUEST.

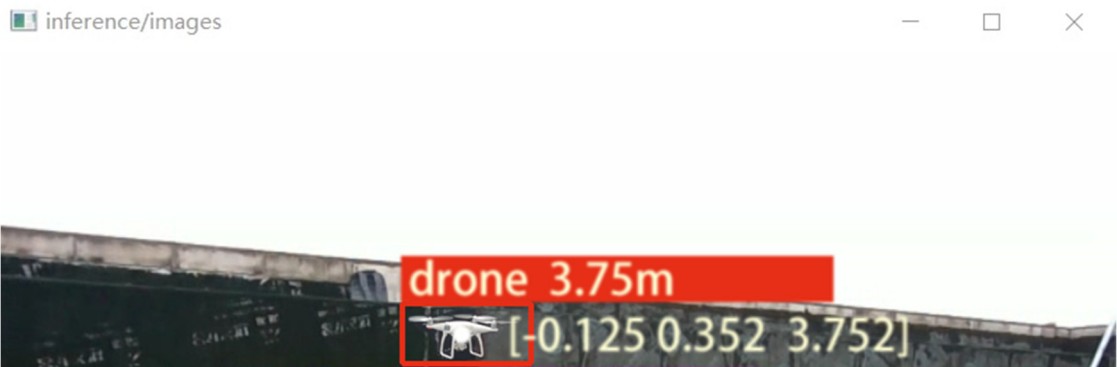

**Figure 9.** Target recognition by YOLO and coordinate calculation.

### 4.3. Result and Analysis

We use the relative attitude angle measured in the experiment as the observation of IF. As mentioned in Section 2.3, the relative attitude angle from the other two nodes could compensate for the error of attitude caused by low-precision INS installed on LAN Node 2.

As for the true angle as the reference, the attitude calculated by INS installed on LAN can represent the true relative angle between Node 2 and HAN Node 1 because the LAN is not moving. Similarly, the true angle between Node 2 and Node 3 should be a fixed value due to no relative motion between the two platforms.

Figure 10a shows the three-axis attitude angle determined by visual vectors. Because we just have the HAN rotated in place on level ground, so the roll and pitch should theoretically be zero. The reference of yaw is shown in (b), which is offered by INS installed, as shown in Figure 7d.

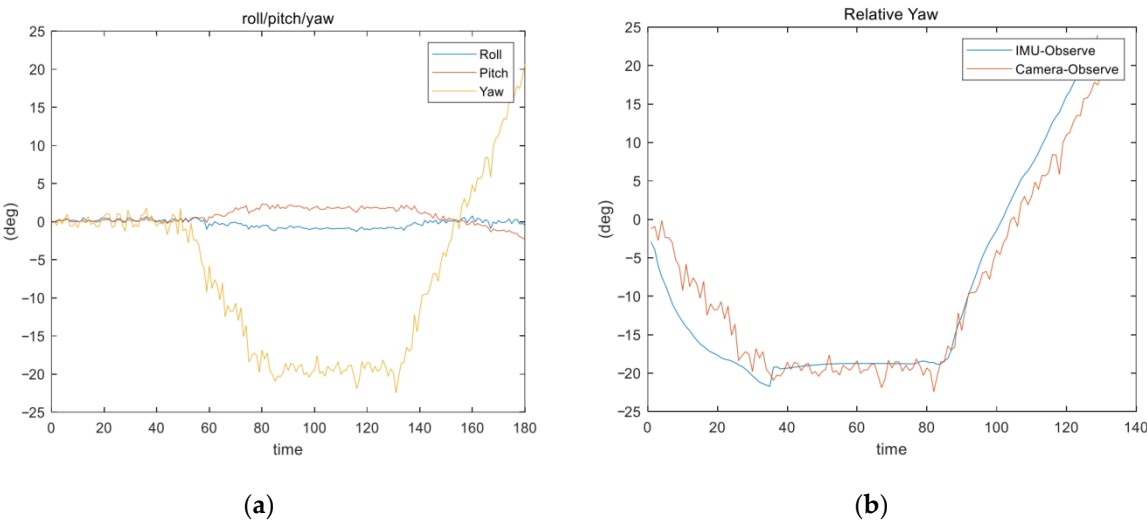

**Figure 10.** The observation result of HAN: (**a**) Roll, pitch and yaw obtained by visual vectors; (**b**) Yaw of the HAN observed by IMU output and by visual vectors during the movement.

The error of angle determined by visual vectors in three-axis is shown in Figure 11. Statistically, the RMS is less than four degrees in the whole moving phase of HAN. When the HAN is stationary, the error is less than one degree but would increase to nearly four degrees when the node starts moving.

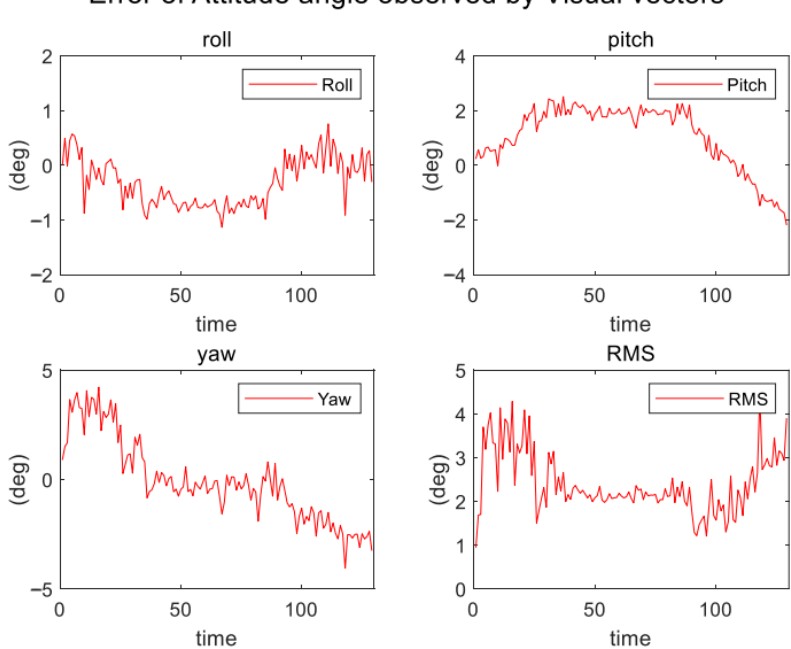

**Figure 11.** Error of attitude angle observed by visual vectors, which act as the observation in the IF.

The main reason for the huge error could be explained as follows:

- Delay caused by calculation
- Inaccurate target recognition

After obtaining the relative angle, we used it as observation in the IF to compensate for the noncommutativity error caused by INS installed on the LAN. The result of IF shows in Figure 12. The errors of angle in three-axis are all less than two degrees after compensation.

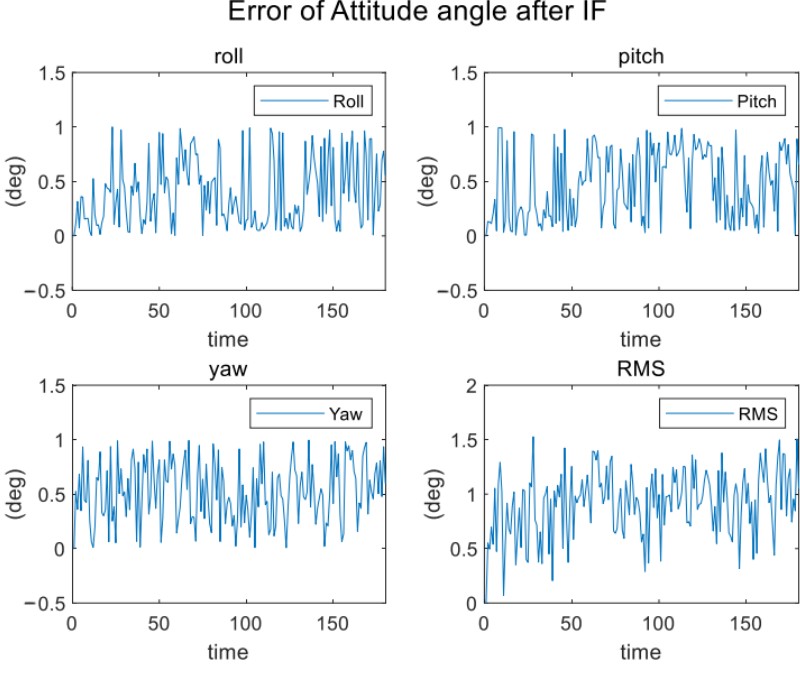

**Figure 12.** Error of attitude angle observed after the compensation of IF.

## 5. Conclusions

In this paper, we design a new method to determine the attitude angle of LANs in unmanned CN swarm based on IF using a relative attitude transformation matrix calculated by high-precision visual vectors. Considering the various limitations of unmanned platforms, we apply easy and cheap sensors to realize the method in practice. Based on verifying the improved QUEST and IF as designed, real-site experiments show that the system has good relative orientation accuracy, comprehensive performance, and application potential.

We carried out experiments using three unmanned platforms in the air and on the ground, each of which installs navigation sensors with different accuracy. In a swarm consisting of three platforms, the LAN among them can compensate for the attitude error according to the attitude angle of the high precision node and the observed relative angle through IF. The three-axis attitude angle error is within three degrees.

However, the research involved in this paper can be improved both in theory and practice. In Section 5.1, we simply summarize and explain the problems reflected in the experimental results. In Section 5.2, we envisage several types of measures that can be improved according to the defects mentioned previously, and give a simple implementation plan and feasibility verification.

### 5.1. Shortage

#### 5.1.1. Ranging Is not Accurate Using Camera

From the experimental results, it can be seen that the precision of LANs' attitude determination depends on the precision of relative attitude transformation matrix, that is, the precision of visual vector measurement. According to the Equation (35):

$$Z_c \begin{bmatrix} u \\ y \\ 1 \end{bmatrix} = \begin{bmatrix} f_x & 0 & u_0 & 0 \\ 0 & f_y & v_0 & 0 \\ 0 & 0 & 1 & 0 \end{bmatrix} \begin{bmatrix} R & T \\ \overrightarrow{0} & 1 \end{bmatrix} \begin{bmatrix} X_w \\ Y_w \\ Z_w \\ 1 \end{bmatrix} \tag{35}$$

The accuracy of the target's 3D coordinates in Frame-C is tightly related to the ranging accuracy, so the ranging error will have a significant impact on the calculation of the attitude angle, especially when visual vectors are not multitudinous. In general, large-ranging errors often occur in the following situations:

- Target occupies too small a proportion in the identification box.
- Target is hollowed out and the shape is irregular.
- Target is a long distance away.

For situation 1 and 2, the target contour can be precisely cut by improving the reduction of target recognition methods, which ensures the points ranged are just within the target object. To solve the problem in situation 3, we just need a camera with a long base line, but it is impracticable in proportion to the size of the platform.

#### 5.1.2. Problems in Identification for Inter Class Targets

In our experiment, the targets we set are all different from each other for YOLO. For most target recognition algorithms, distinguishing inter class targets requires additional samples and time for training.

### 5.2. Future Improvements

#### 5.2.1. Target Recognition Based on Semantic Segmentation

Image Semantic Segmentation obtains the content-based annotation of an image by analyzing the content of the training image, and at the same time, it obtains the semantic categories of all the segmentation areas and even each pixel of the image. Image semantic segmentation requires not only accurate identification of the boundary of the image segmentation area, but also accurate identification of the target categories of the segmentation area [25].

Compared to other object detection methods such as YOLO or R-CNN, employing Image Semantic Segmentation could ensure that the pixel points ranged are exactly on the targets detected, which makes the ranging result free of the interference from the background.

### 5.2.2. Joint Probabilistic Data Association for Identification of Unclassified Targets

Joint probabilistic data association (JPDA) is one of the data association algorithms [26]. The purpose of JPDA is to calculate the correlation probability between observation data and each target, and it is believed that all effective echoes may originate from each specific target, but their probabilities from different targets are different. The advantage of the JPDA algorithm is that it does not need any prior information about the target and clutter, and so it is one of the better methods to track multiple targets in a cluttered environment. Therefore, JPDA could identify the unclassified targets according to their unique trajectory. However, when the number of targets and measurements increases, the computational load of JPDA algorithms will explode, resulting in computational complexity.

### 6. Patents

ZL 2022 1 0883012.8.

**Author Contributions:** Conceptualization, Y.L. and R.L.; methodology, Y.L.; software, Y.L. and R.L.; validation, Y.L. and R.L.; formal analysis, P.J.; investigation, Y.G.; resources, Z.X.; data curation, S.C.; writing—original draft preparation, Y.L.; writing—review and editing, Y.L.; supervision, R.Y.; project administration, Y.L. and R.Y.; funding acquisition, R.Y. and Z.X. All authors have read and agreed to the published version of the manuscript.

**Funding:** The APC was funded by Natural Science Foundation of China: 62203456, 42276199.

**Data Availability Statement:** Data sharing not applicable. No new data were created or analyzed in this study. Data sharing is not applicable to this article.

**Acknowledgments:** Thanks to the help and hardware support from Doctor Jinyi Yang and other teachers in the Navigation and Guidance Laboratory.

**Conflicts of Interest:** The authors declare no conflict of interest.

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
