# Peer review of "Attitude Determination for Unmanned Cooperative Navigation Swarm Based on Multivectors in Covisibility Graph"

_drones, doi:10.3390/drones7010040_

Round 1

Reviewer 1 Report

The presented article is definitely interesting, but the negligence of the authors spoils the whole impression.

1. The article contains many spelling errors. Editorial processing required.

2. The first use of the terms must be deciphered.

3. Abundance of inaccuracies in the text (Frame-N 171, Frame-C, B without explanation), 410-424 non-deleted article template lines, lines 379, 370, 372, bibliography style, etc.

4. Line 427 states the verification of the algorithm. However, the algorithm is not presented by the authors.

5. Line 433 sets obscure values and does not explain the choice of values. The experiment is not justified - the choice of the platform and the number of tests is not clear.

6. No comparison with other methods.

7. In line 539, the authors themselves acknowledge the lack of complexity in the solution.

8. A number of paragraphs, for example, 303, 342, duplicate the introduction.

9. For formulas 18-19: Lambda indexes is never used, Eq18 has numeric coefficients with no explanation, Recursive Cyrillic G is not obviously calculated.

10. The authors use the visual component, which is very sensitive to the environment, and do not consider alternative ways to solve the problem.

Reviewer 2 Report

Very thorough paper. The derivation is accompanied by good descriptions/explanations and is demonstrated with both simulation and experimental examples. However, there are numerous grammatical errors throughout the draft that distract from the scientific content. Additionally, lines 410-424 immediately before section 3 appear to be from the paper template and should be removed.  

In section 3.1, a short description of the six test cases used from Markely would be useful context.

A variable is missing on the left-hand side of the equation in line 433.

(suggestion) In Tables 1-3, color-coding the values based on their relative magnitude would accelerate visual comprehension of the results. 

Author Response

Point 1:There are numerous grammatical errors throughout the draft that distract from the scientific content. Additionally, lines 410-424 immediately before section 3 appear to be from the paper template.

Response 1: I feel very guilty about the poor grammar and errors in the article. Honestly, the writing process of the article is very hasty. I have checked the article carefully and corrected most of the errors.

Point 2:  In section 3.1, a short description of the six test cases used from Markely would be useful context.

Response 2: The source and reason for these 6 sets have been added.

Point 3:  A variable is missing on the left-hand side of the equation in line 433.

Response 3:The missing has been added.

Point 4:  (suggestion) In Tables 1-3, color-coding the values based on their relative magnitude would accelerate visual comprehension of the results.

Response 4: For these 3 tables, the part filled with blue means performs better than the algorithm we newly proposed. Similarly, green and red parts represent equal and worse performance respectively.

Round 2

Reviewer 1 Report

Authors done good work at reviewers' comments but I see no changes in positions 426-440 (only red color) and References. Other questions was closed by responses in the article and authors' comments

Author Response

1. Redundant text in line 426-440 have been deleted now.

2. Mistakes of bibliography pattern have been corrected again. The new added reference is numbered as reference [26] and write in blue now.